# Early Adverse Reactions to Snake Antivenom: Poison Center Data Analysis

**DOI:** 10.3390/toxins14100694

**Published:** 2022-10-09

**Authors:** Charuwan Sriapha, Panee Rittilert, Taksa Vasaruchapong, Sahaphume Srisuma, Winai Wananukul, Satariya Trakulsrichai

**Affiliations:** 1Ramathibodi Poison Center, Faculty of Medicine Ramathibodi Hospital, Mahidol University, Bangkok 10400, Thailand; 2Snake Farm, Queen Saovabha Memorial Institute, The Thai Red Cross Society, Bangkok 10330, Thailand; 3Department of Medicine, Faculty of Medicine Ramathibodi Hospital, Mahidol University, Bangkok 10400, Thailand; 4Department of Emergency Medicine, Faculty of Medicine Ramathibodi Hospital, Mahidol University, Bangkok 10400, Thailand

**Keywords:** incidence of early adverse reactions, early adverse reaction rate, snake antivenin, adverse reactions, anaphylactic reactions, snakebite, snake venom

## Abstract

Antivenom is an essential treatment for snake envenomation; however, early adverse reactions (EARs) are major limitations to its use. We performed a retrospective cross-sectional study using Ramathibodi Poison Center data (January 2016 to December 2017) to clarify the incidence and severity of EARs following different F(ab’)_2_ antivenoms. Among 1006 envenomed patients, 684 (68%) received antivenom therapy with a total of 1157 doses, mostly green pit viper antivenom. The overall EAR incidence and rate were 22. 5% (154/684) and 15% (173/1157), respectively. The EAR rate following each type of antivenom was >10%, except for Russell’s viper antivenom (2.9%); the severe reaction rate was 2.6% (30/1157). Malayan pit viper bites caused a high incidence of EARs (37.8%) and the highest EAR rate (22.3%). Fifty-two cases developed anaphylaxis. All EARs occurred within 2 h after treatment initiation. No deaths were attributed to EARs. The duration of administration was significantly different between doses of antivenom that induced EARs and those that did not. In conclusion, all types and every dose of antivenom should be infused for 30–60 min. Preparation of resuscitation equipment and continuous clinical observation are crucial for at least 2 h after administration, and prompt treatment should be provided when EARs occur.

## 1. Introduction

Snakebite envenomation is the cause of enormous amounts of suffering, disability, and death in many parts of the world [1,2], but the highest number of snakebite victims are reported in sub-Saharan Africa and South and Southeast Asia [2]. In Thailand, snakebite envenomation is prevalent and is an important medical emergency that causes morbidity and mortality [1,2,3,4,5]. The average annual incidence in Thai people is estimated to be 5035 bites [6]. The incidence has increased from 1753 bites (2.66 per 100,000 inhabitants) in 2016 to 4203 (6.36 per 100,000) in 2017 and 7247 (10.93 per 100,000) in 2018. In addition, the mortality rate has escalated from 0% in 2014 to 0.32% in 2018 [6]. The World Health Organization added snakebite envenoming to its priority list of a highly neglected tropical diseases and aims to reduce the number of deaths and cases of disability by 50% before 2030 [1]. The neglect of snakebite envenoming includes a lack of access to life-saving antivenom treatments. Antivenoms are preparations of immunoglobulins purified from the plasma of a hyperimmunized animal [7]. The parenteral administration of specific antivenoms is an essential treatment element to prevent or reverse most of the harmful effects of venomous snakebites [7]. However, patients can develop early adverse reactions (EARs) to antivenom following therapy. EARs to antivenom comprise a group of adverse manifestations that occur within 24 h of administration of the antivenom [8]. The risk of EARs is a major limitation of antivenom use, especially as some EARs, such as anaphylactic shock, may be life-threatening [9,10].

The number of reported EARs to snake antivenom varies, with the incidence of EARs ranging from as low as 3% [11] to as high as 88% [12]. Such high variability, which makes it difficult to compare studies, may result from many factors, including the different types or purities of antivenoms used and differences in treatment protocols and populations.

There are at least 46 manufacturers of animal-derived antivenoms worldwide [13], one of which is the Queen Saovabha Memorial Institute (QSMI) of The Thai Red Cross Society, the only antivenom manufacturer in Thailand since 1923 [14]. To date, QSMI produces both monovalent and polyvalent antivenoms for medically important snakes in Thailand. Snakebite incidents are no longer restricted to tropical and subtropical countries because of environmental and human factors; for instance, climate change, rapid urbanization, and the growing global trade in non-native snakes [15,16]. Therefore, QSMI antivenoms are not only used throughout Thailand but also exported to save the lives of patients bitten by specific types of venomous snakes or closely related species in other countries [14], such as Laos [17], Hong Kong [18], and the United Kingdom [19,20]. This means that several QSMI antivenoms are distributed, stocked, and used globally, and they are regarded as some of the most effective antivenoms in the world [21].

However, there remains a significant amount of controversy over the frequency and severity of adverse reactions or anaphylaxis following antivenoms. Previous studies reported incidences of EARs to QSMI antivenoms ranging from 0% to 53%, with incidence of severe reactions ranging from 0% to 30% [11,17,22]. These disparate reports are needed to assess the adverse events from these antivenoms and to ensure the best possible treatments are available for snakebite victims.

Many patients who have incurred a snakebite in Thailand have consulted or been referred to the Ramathibodi Poison Center (RPC). The safety of antivenom is an important concern; however, there are a limited amount of data on EARs for the individual antivenoms available for different snake species. Accordingly, this study was undertaken to clarify the incidence and severity of EARs among different types of antivenoms and the factors associated with EARs to the snake F(ab’)_2_ antivenoms produced by QSMI. This study aimed to expand our knowledge of EARs occurring from antivenom use.

## 2. Methods

### 2.1. Study Design

A retrospective cross-sectional study was performed using data collected from the RPC Toxic Exposure Surveillance System, which is our poison center database, from January 2016 to December 2017. The primary outcomes were incidence of EARs and EAR rate from all types and each type of snake F(ab’)_2_ antivenom produced by the QSMI, Thailand. The secondary outcome was factors associated with EARs.

### 2.2. Study Site and Population

The RPC is a poison center based in a tertiary teaching hospital that serves the entire country with a 24-h telephone service for both healthcare professionals and the general public. Most queries to the RPC are from medical personnel, and there are approximately 20,000–30,000 consultations annually. Followup calls are made to collect data on patients, monitor case progress, and determine the medical outcomes of patients. All cases are recorded in the RPC Toxic Exposure Surveillance System and verified by a team of senior information scientists and medical toxicologists. All patients with a history of venomous snakebites who received antivenoms from January 2016 to December 2017 were enrolled in the study. The exclusion criteria included patients with snake venom ophthalmia.

The medically important snakes in Thailand are toxic primarily to either the neuromuscular or hematologic systems. Neurotoxicity is typically caused by elapid snakes, namely the king cobra (KC; *Ophiophagus hannah*), cobra (*Naja* species), banded krait (BK; *Bungarus fasciatus*), and Malayan krait (MK; *B. candidus*). Hematotoxicity is induced by venom from the green pit viper (GPV; *Trimeresurus* species), Malayan pit viper (MPV; *Calloselasma rhodostoma*), and Russell’s viper (RV; *Daboia siamensis*). Other venomous snakes also inhabit Thailand [23,24]; however, their bites are rarely reported to cause morbidity, disability, or death [7]. These snakes include *B. flaviceps*, *B. slowinskii*, *B. wanghaotingi*, *Naja sumatrana*, *Ovophis* species, *Protobothrops* species, and Asian coral snakes [23,24].

The species of snake responsible for envenoming was determined via the following: (1) Identification of the snake carcass or a photograph that the patient or relative brought along to the hospital or sent to the RPC’s LINE application. All identities were confirmed by the RPC’s consultant, an author who is a highly experienced professional veterinarian from Snake Farm, QSMI; (2) The patient’s or relative’s description of the morphology of the snake and/or the snake’s species name. Because this information is sometimes unreliable, snake identification relied on the patient’s recognition corresponding to the area of distribution/habitat of the snake species, the patient’s local and systemic clinical features of envenomation such as local swelling or coagulopathy, laboratory abnormalities, and the patient’s response to a specific antivenom; (3) In cases for which the snake was seen, but could not be identified by the victim (undetermined), the final diagnosis was based on the presence of fang marks that were consistent with the patient’s local and systemic clinical features of envenomation, laboratory abnormalities, together with their response to a specific antivenom. If the type of the snake could not be determined, it was classified as an “unknown hematotoxic snake” or “unknown neurotoxic snake” depending on the clinical symptoms of envenomation and the patient’s response to a specific type of polyvalent antivenom. Additionally, when the patient presented with isolated fang marks without systemic envenomation; it was classified as an “unidentified venomous snake”.

### 2.3. Study Protocol

For snakebite patients who met the inclusion criteria, specific information for analysis, including the patients’ demographic information, medical history, snake identification, type of antivenom given, quantity of antivenoms per dose, number of doses given per patient, premedication drugs, adverse reactions to antivenom, and treatment modalities, were collected.

Snake antivenoms from the QSMI, The Thai Red Cross in Bangkok, Thailand, were used for treatment. The competence of this institute in biological research and production, especially antivenom production, has led to it being acknowledged as a WHO Collaborating Center for Venomous Snake Toxicology and Research. From 2003 to date, QSMI antivenoms have been produced from the plasma of hyperimmune horses challenged with relevant snake venoms. After the plasma is purified, it is subjected to pepsin digestion to cleave IgG into F(ab’)_2_ fragments, and the Fc fragments and unwanted proteins are precipitated by caprylic acid. The F(ab’)_2_ solution is then purified and concentrated by ultrafiltration [25], formulated, and sterilized before filling and lyophilization [26]. These antivenoms are freeze-dried powder with a 5-year shelf life. Monovalent antivenoms are effective against one species or genus of snakes, whereas polyvalent antivenoms are produced via the immunization of one horse with multiple venoms to avoid the high-protein loading that can occur with mixtures of species-specific monovalent antivenoms; therefore, they are able to neutralize toxins from several snake species. There are nine types of products in total and seven types of monovalent antivenoms: those for cobra, KC, BK, MK, GPV, MPV, and RV. The remaining two types are polyvalent antivenoms for hematotoxic (HPAV) and neurotoxic (NPAV) snakes. HPAV is produced to treat GPV, MPV, and RV bites; while NPAV is effective against cobra, KC, MK, and BK venom [21,26].

Antivenom therapy is recommended for systemic envenoming based on the recommendations of the QSMI [27] and The Thai Society of Clinical Toxicology [28]. Hematotoxic envenoming is indicated by one of the following: (1) clinical signs of bleeding; (2) a platelet count of less than 50,000/µL; (3) a venous clotting time of more than 20 min, unclotted 20-min whole blood clotting time, or an international normalized ratio of more than 1.2; and (4) severe pain and swelling due to suspected compartment syndrome. For neurotoxic envenoming, indications for antivenom administration include the following: (1) any clinical symptom or sign of muscle paralysis and (2) definite cases or a suspicion of BK or MK bite, irrespective of symptoms. Whenever the snake species is unknown, our staff uses all patients’ information to narrow the treatment down to only one type of monovalent antivenom. If the patient presents with either hematotoxic or neurotoxic signs of envenoming, but the identification of the biting species is uncertain or was unclear from the clinical symptoms, or monovalent antivenom was unavailable, polyvalent antivenom is preferable. However, because KC and BK antivenoms are not included in the Thai national antidote program [29], and in cases when these antivenoms are not available in a particular hospital, NPAV is used instead.

The appropriate use of antivenom was defined as its administration based on the indications mentioned above. One dose of antivenom for hematotoxic envenoming is 3–5 vials, and for neurotoxic envenoming, it is 5–10 vials. An additional dose is provided as needed every 6 h after the initial dose for hematotoxic envenoming or 12 h after the initial dose for neurotoxic envenoming to gain initial control of envenomation [28], and the regimen can be adjusted according to the patient’ s condition and/or laboratory results [27,28]. Before administration, it is recommended that the antivenom is reconstituted in 10 mL sterile water, diluted in normal saline, and each dose infused intravenously over 30–60 min [30]. For patients with antivenom hypersensitivity, our center advises withholding antivenom infusion, along with resuscitation and symptomatic treatment, treatment for an anaphylactic reaction with oxygen, intravenous (IV) fluids, and parenteral route, particularly intramuscular (IM), adrenaline. Premedication is then considered if antivenom is still indicated, and antivenom is recommenced at a slower infusion rate when symptoms have resolved. However, the decisions concerning treatment depend on the physician supervising the patients.

The EARs are classified using the Brown Grading System [31] as none, mild (skin and subcutaneous tissue changes, such as generalized erythema, rash, periorbital edema, or angioedema); moderate (features suggesting respiratory, cardiovascular, or gastrointestinal involvement, including dyspnea, stridor, wheezing, nausea, vomiting, dizziness, diaphoresis, chest or throat tightness, or abdominal pain); and severe (hypoxia, hypotension, or neurological compromises, including cyanosis, oxygen saturation < 90%, confusion, collapse, or loss of consciousness) reactions, based on the details recorded for each adverse reaction in the RPC database. The definition of anaphylaxis, according to this grading system, is an EAR that correlates closely with the World Allergy Organization criteria of anaphylaxis [32]. Every dose of antivenom that induced adverse reactions was analyzed in this study.

After collating the data on patients who developed EARs, the incidence of EARs was calculated as the number of patients with EARs divided by the total number of patients who received antivenom and multiplied by 100.
(1)$$Incidence\;of\;EARs=\frac{Number\;of\;patients\;with\;EARs}{Total\;patients\;receiving\;antivenom}\;\times \;100$$

To help clarify the exact type of antivenom that was related to a high rate of adverse reactions and clinical severity, we further separated each dose of antivenom that each patient received and identified which dose or type of antivenom was related to EARs and calculated the EAR rate. Therefore, after determining the doses of antivenom that caused EARs, the EAR rate was calculated from the number of doses of antivenom causing EARs divided by the total doses of antivenom used and multiplied by 100.
(2)$$EAR\;rate\;of\;given\;antivenom\;doses=\frac{Number\;of\;antivenom\;doses\;causing\;EARs}{Total\;doses\;of\;antivenom\;given}\;\times \;100$$

### 2.4. Statistical Analysis

The RPC data were obtained and managed using Microsoft Excel, then analyzed with IBM SPSS Statistics for Windows, version 18 (IBM Corp., Armonk, NY, USA). Descriptive statistics were performed to characterize the data. The mean, median, minimum, maximum, standard deviation, and interquartile range (IQR) were analyzed for continuous data, and frequency and percentage were analyzed for categorical data. Between-group comparisons were performed by Student’s *t*-test if the data were normally distributed, and by the Mann-Whitney U test otherwise. Differences in categorical variables were evaluated by chi squared analysis and Fisher’s exact test. A *p*-value of 0.05 or less was considered statistically significant.

## 3. Results

### 3.1. Characteristics of Snakebite-Envenomed Patients and Antivenom Therapy

The groups analyzed in the study are outlined in Figure 1. There were 46,241 cases of poisoning and toxic exposure during the 2-year study period. Among these, 1287 patients were registered as having snakebites. After 281 cases were excluded because they had nonvenomous snakebites; 1006 venomous snakebite cases were included in the analysis. The median age of patients was 38 years (range 0.6–97), and snakebites predominantly affected males (60%). Information on the snakes identified is listed in Figure 1. We discovered that 2.0% (20 cases) of all patients who provided the snake’s common name provided an identification that was inconsistent with the clinical syndrome of envenomation and their response to the specific antivenom. The biting snake species was identified in 85.8% of cases, about half of which (42.5%) were identified by the expert, as presented in Table 1. The patients were mainly bitten by hematotoxic snakes (595 cases, 59.1%), and the most common species of biting snakes were GPV (336 patients, 33.4%) and MPV (168 patients, 16.7%). In neurotoxic envenomation cases (356 cases, 35.4%), most bites were from cobras (271 patients, 26.9%). The type of venomous snake was unidentified for 55 patients (5. 5%) who had fang marks at the bite site and were suspected of having a venomous snakebite but did not develop systemic symptoms that were suggestive of the responsible species. Among the 1006 included patients, 684 (68%) received antivenom therapy. The median age of these patients was 38 years (range 1–97), and most were male (61.1%) (Table 2). Of these, 429 patients (62.7%) were treated for hematotoxic snakebites and 255 (37.3%) for neurotoxic snakebites. The three most common species of snakes implicated were GPV (213 patients, 31.1%), cobra (188 patients, 27.5%), and MPV (143 patients, 20.9%). The total number of antivenom doses administered was 1157 (13,672 vials). Most cases (95.2%) received one type of antivenom throughout their clinical management. The median number of antivenom treatments was 1 dose (IQR 1–2, range 1–15) or 5 vials (IQR 3–9, range 1–75) per patient with hematotoxic envenomation and 1 dose (IQR 1–1, range 1–4) or 10 vials (IQR 10–10, range 1–30) per patient with neurotoxic envenomation. The overall mortality rate of patients who received antivenom therapy was 1.75% (12/684 patients), including patients with cobra bite (6 patients), KC bite (2 patients), MK bite (1 patient), MPV bite (1 patient), RV bite (1 patient), and unknown hematotoxic snakebite (1 patient). 

A comparison of the median number of antivenom doses received by each patient with hematotoxic envenomation (mean = 1 dose, range 1–15 doses) and those with neurotoxic envenomation (mean = 1 dose, range 1–4 doses) demonstrated a statistically significant difference (*p* < 0.001).

The premedication drugs given to patients before antivenom administration by the treating physicians were varying combinations of antihistamines, corticosteroids, or adrenaline. Thirty-nine (5.7%) of the 684 patients were given pretreatment drugs, which equated to 58 (5.0%) of the 1157 doses of antivenom. These drugs included chlorpheniramine IV (23 doses); chlorpheniramine IV, and hydrocortisone IV (1 dose); chlorpheniramine IV and dexamethasone IV (30 doses); chlorpheniramine IV, dexamethasone IV, and adrenaline IM (3 doses); and adrenaline IM (1 dose). Of these, 8 patients (16 doses) still had adverse reactions to the antivenom.

### 3.2. Incidences of Early Adverse Reactions to Antivenom Per Patient

Of the 684 patients who received antivenom, 154 developed EARs, most of whom were male (65.6%), with a median age of 35 years (range 1–87). The total number of doses of antivenom administered was 259 (1223 vials), of which 173 doses (66.8%) induced EARs, and the overall incidence of EARs was 22.5% (Table 2). There was a higher incidence of EARs in patients with hematotoxic snakebites (25.2%, 108/429 patients) compared those with neurotoxic snakebites (18%, 46/255 patients). In hematotoxic envenomation cases, the highest incidence of EARs to antivenom was reported for MPV (37.8%) bites, followed by GPV (20.7%) bites, and the median number of doses that induced EARs per patient was 2 (IQR 1–2.75, range 1–6) or 5.5 vials (IQR 3–9, range 1–18). The highest incidence of EARs in neurotoxic envenomation cases occurred for BK (50.0%) bites, followed by KC (42.9%) bites. However, the number of patients bitten by these two types of snakes was relatively low. The reliable incidences of MK and cobra bites were 20% and 17%, respectively, because the number of patients bitten by these species was higher. The median number of antivenom doses that induced EARs per patient was 1 dose (IQR 1–1, range 1–4) or 10 vials (IQR 10–10, range 1–24).

Most of the cases (145 patients, 94.2%) received one type of antivenom, and the EARs incidence was 22.3%. The remaining 9 (5.8%) patients received two types of antivenom, for which the incidence of EARs increased to 29.0%. The most common combination of antivenoms was GPV and HPAV (4 of 9 patients). Two patients were given MPV and HPAV antivenoms, one patient MPV and GPV antivenoms, one KC and NPAV antivenoms, and one cobra and NPAV antivenoms. No patient receiving three types of antivenom developed any reactions; however, the sample size was too small (2 patients) to determine the significance of this.

When we compared the patients that received antivenom with and without the presence of EARs according to age, gender, median number of antivenoms (vials) received per dose, median of the total number of antivenoms (vials) received per patient, neurotoxic envenomation, and type of neurotoxic snakebite, there were no statistically significant differences in those parameters. However, a comparison of patients with hematotoxic snakebites with those with neurotoxic envenomation and the different types of hematotoxic snakebites among patients with hematotoxic envenomation demonstrated statistically significant differences between the two groups of patients with *p*-values of 0.031 and 0.001, respectively.

### 3.3. Early Adverse Reaction Rate Per Dose of Each Antivenom

The types and quantities of antivenoms given are presented in Table 3. Eight of the nine types of antivenoms produced by QSMI were administered; BK antivenom was not used because the two patients with BK bites received NPAV. The 3 most common types of antivenoms administered were GPV (341 doses, 3860 vials), MPV (283 doses, 3071 vials), and cobra (213 doses, 2754 vials) antivenoms, accounting for 72.5% of all antivenom doses. Of the patients who presented EARs, 154 of them received a total of 259 doses of antivenom, but only 173 (66.8%) doses induced adverse reactions. As illustrated in Figure 2, 88 (57.1%) patients who received a total of 1 dose each presented reactions at an EAR rate of 20.7%. For the 37 (24.0%) patients treated with a total of 2 doses, the reported EAR rate was 21% at the first dose, 13% at the second dose, and 3.6% after receiving both doses. The EAR rates for the remaining patients are demonstrated in Figure 2, including 21 patients who received a total of 3 doses, 7 patients who received 4 doses, and 1 patient who received 6 doses. However, in a patient treated with a total of 6 doses, only the sixth dose resulted in an EAR. Interestingly, no EARs were observed in patients who received a total of 5 doses. Moreover, a patient who received 15 antivenom doses in total never developed EARs.

The overall EAR rate for all types of causative antivenoms was 15%. For hematotoxic antivenoms, MPV antivenom involved the highest EAR rate of 22.3%, followed by HPAV (15.1%) and GPV antivenom (11.4%). In cases receiving neurotoxic antivenoms, the highest rate was from NPAV (16.4%), whereas cobra and KC antivenoms were implied in EARs with rates of 14.6% and 14.3%, respectively. Three-fourths of the doses (131 doses, 75.7%) induced EARs during the first hour after initiation of antivenom treatment or during administration, with a median time of onset of 15 min (IQR 10–30, range 1–60). The remaining doses (42 doses, 24.3%) incurred EARs during the first hour after administration was completed, and the median time of onset of reactions was 5 min (IQR 5–45, range 2–60).

## 4. Clinical Features, Severity Grading, and Treatment of Patients with EARs

The patients who developed EARs from the 173 doses of antivenom presented the following clinical reactions: skin rash (68.8%), chest tightness (24.3%), dyspnea (16.2%), bronchospasm (16.2%), and hypotension (15.0%). The other presentations are depicted in Figure 3. From the overall EAR rate of 15.0%, the severity grading of EARs was categorized as mild for 6.1%, moderate for 6.3%, and severe for 2.6% (Table 3). KC antivenom portrayed the highest percentage (14.3%) of severe reactions; however, the causative dose number was too small to determine the significance (1 out of 7 doses). Severe EARs were most observed after MPV (5.3%), followed by NPAV (4.1%), HPAV (3.4%), RV (2.9%), cobra (1.4%), and GPV (0.3%) antivenoms. We did not find records of severe reactions after MK antivenom administration. Notably, most severe EARs (25/30 doses, 83.8%) developed with a median time of onset of 10 min (IQR 5–15 min; range 2–60 min) after the initiation of antivenom treatment.

Of the 154 patients in total with EARs, 52 cases (38%) were classified as having anaphylaxis according to World Allergy Organization criteria. Of these, five patients received premedication, four were given intravenous chlorpheniramine and dexamethasone, and one received intravenous chlorpheniramine only. Anaphylaxis with severe features developed in 30 patients, and symptoms included hypotension (26 patients), cyanosis (5 patients), and loss of consciousness (1 patient). These reactions were treated mostly with fluid replacement, followed by intravenous chlorpheniramine (40.2%) and intravenous dexamethasone or hydrocortisone (28.2%). Other treatment modalities included intramuscular or intravenous adrenaline (18.0%), intravenous ranitidine (5.7%), intravenous fluid loading (3.2%), bronchodilator nebulizer (2.5%), oxygen therapy (1.9%), and intravenous dopamine (0.3%). All clinical reactions responded well to treatment. There were no deaths attributed to EARs, and no cases of serum sickness or delayed antivenom reaction were reported in our patients.

### Factors Associated with EARs

To determine the factors that might be associated with EARs, we further compared the characteristics of antivenom administration (each dose) that induced EARs with those of the antivenom doses that did not (Table 4). The only factor demonstrating a statistically significant difference between the two groups was the duration of antivenom administration (*p* = 0.002), and an infusion over 30–60 min—as was mentioned as the recommended method in this report—was associated with a significantly lower incidence of EARs than those of <30 min and those of >60 min. Consequently, we compared the duration of antivenom administration using only 2 groups, a recommended group (30–60 min) and a nonrecommended group (<30 and >60 min). We discovered that the recommended infusion rate had significantly lower incidence of EARs than the nonrecommended group (*p* < 0.001). Additionally, the different types of antivenom for hematotoxic snakes were significantly associated with EARs (*p* = 0.001). However, the different types of antivenom for neurotoxic snakes administered to the patients were not associated with EARs (*p* = 0.936).

## 5. Discussion

The correct identification of the venomous snake responsible for a bite is crucial for optimal clinical management of the victim, particularly when selecting a specific and appropriate antivenom [1,7]. In our study, the snake species was identified for most of the patients, and half of the identifications were confirmed by an expert. These professional identifications helped to confirm the type of snake and the correct choice of antivenom in our patients, as snakes can be incorrectly identified by laypersons [7]. This supports our finding that 2.0% of the snakes identified from the description given by the patient and other witnesses did not correspond to the clinical syndrome of envenomation. Venomous snakes of Thailand are generally toxic to either the hematologic or neuromuscular systems [30]. Although, analysis of the specific toxin is not generally available or is usually reported too late for clinical decision-making. The context and specific information, including the patient’s description, the circumstances of the bite, clinical manifestations, and/or the response to a specific antivenom, are helpful as auxiliary means of snake identification and for diagnosing the correct envenomation to guide proper management [7,30,33]. Venomous snakebites, which are an important health problem in the Thai population, are predominantly caused by GPV, followed by cobras and MPV, for which antivenom treatment is required. Epidemiological data reported before 2015 indicated that most venomous bites in Thailand were due to MPV [6]. However, our findings, in addition to those of studies from 2016 onward, demonstrate a changing trend, with *Trimeresurus* species or GPV being the most common type [3,6]. A higher risk of snakebites was found in adult males than females, and the peak incidence was in young adults. This finding corroborates previous studies that demonstrated young adult males were more likely to be afflicted with snakebites than females, primarily because of their agricultural activities [6,7].

During our study period, eight of the antivenom types produced by QSMI were administered to 68% of all snakebite patients. Of these, 154 patients with EARs were further analyzed. A total of 684 envenomed patients received antivenom, and to our knowledge, this is the only large study of both the incidence and rate of EARs to snake antivenom in the English scientific literature, particularly EARs due to antivenoms composed of F(ab’)_2_ fragments.

Adverse reactions to snake antivenoms can be both early-onset and late-onset [8]. This retrospective study described the instances of snakebite envenomation and EARs after the administration of QSMI antivenoms. EARs are classified into pyrogenic and anaphylactic reactions [7,8,34]. Pyrogenic reactions are usually caused by contamination with antivenoms with pyrogenic substances but are rarely reported because pyrogen testing is a routine quality control procedure implemented by manufacturers [7], which is why no pyrogenic reactions, such as fever, were observed in our study. The mechanisms involved in most EARs have not been clearly determined [7,35]. Anaphylactic reactions are divided into those caused by mechanisms involving IgE-mediated and non-IgE-mediated reactions. Most EARs occurring after antivenom administration are attributed to non-IgE-mediated reactions, such as complement activation by IgG aggregates or residual Fc fragments, or the stimulation of mast cells or basophils by antivenom proteins [7,36]. In previous studies, cases of antivenom reaction were demonstrated to not be complement-mediated but were more likely due to IgG immunoglobulin complexes or impurities in the antivenom [34,36]. The predominance of non-IgE-mediated reactions is supported by the finding in our study that some patients who developed EARs were able to receive subsequent doses of the same antivenom without incurring further EARs. However, the exact mechanisms involved in EARs to antivenom are still not clearly understood [7,8,34]. Although they can be instigated via various mechanisms; most EARs portray similar clinical manifestations; therefore, a prior incidence of EARs to antivenom is not an absolute contraindication to the subsequent administration of antivenom.

EARs, including anaphylactic reactions, in our study occurred in nearly one-quarter (22.5%) of patients administered antivenom, irrespective of snakebite species or type of QSMI antivenom used. A higher incidence was observed in patients with hematotoxic envenomation (25.2%) than neurotoxic envenomations (18.0%). It is difficult to explain and form definite conclusions on the relationships between the reactions to antivenom and envenomation due to these two groups of snake species. Possible reasons are the variations in some constituents of snake venom [7,37] or the antigenicity involved in inducing specific neutralizing antibodies during the manufacture of different snake species antivenoms [7,38]; or physiochemical differences, such as the heterogeneity of proteins, among the antivenoms [7,8]. The different types of snakebite and/or antivenoms received might result in different immune responses to immunoglobulin, differences in the antigenicity of the antivenom, or the reaction between venom and antivenom. Our other proposed explanation is that this finding might be attributable to the fact that patients bitten by hematotoxic snakes required more repeated antivenom doses than patients bitten by neurotoxic snakes (*p* < 0.001). The repeated administration of antivenom would have led to their exposure to more of the heterologous proteins or antigens in the antivenom. Moreover, it is unknown how complexes composed of F(ab’)_2_ and venom are eliminated [38]. If the elimination or neutralization is incomplete, these complexes might be capable of eliciting the EARs observed more frequently in hematotoxic snakebite patients. Interestingly, when we compared the type of hematotoxic and neurotoxic antivenom administered with EAR occurrence, calculated by the number of doses of antivenom (Table 4), they were not significantly different. This might be because some doses of antivenom were given to patients sequentially, not freely administered as one dose for one patient, so the sequence of the dose or repeated dose or dose accumulation might have affected the EAR rate in our patients. However, there was no statistical association between the number of antivenom doses and the presence of EARs in all patients. This is distinctly different from previous studies, which showed that the risk of a reaction to antivenom was dose-related [7,11,39]. However, our finding is similar to that described by Deshpande et al. [40], who reported more antivenom reactions in hematotoxic snakebite patients (52.2%) than neurotoxic ones (21.7%), and the reactions were not dose- dependent [34]. One study also reported no correlation between antivenom dose and the development of EARs [41]. Another study performed in South Africa described that there was no association between antivenom doses and the anaphylaxis following the administration of the South African Vaccine Producers antivenom [42].

In the present study, incidences of EARs to antivenom varied among the types of biting snake. The highest incidences of reactions were seen for BK and KC bites, but further studies are required to confirm the results, as the number of patients bitten by these two neurotoxic snakes was very low. Notably, patients bitten by BK were treated with NPAV, not BK antivenom, which our results implied was associated with a high EAR rate and more severe EARs. Patients who received KC antivenom developed EARs with the highest rate of severe reactions (14.3%). Our findings concur with a report of a 22-year-old man in the United Kingdom [18] being bitten by a KC and developing severe anaphylaxis after receiving KC antivenom. Based on this information, for both KC and BK bites, if antivenom is indicated, it should be administered with caution in a critical care setting equipped to treat anaphylaxis.

Among the patients bitten by hematotoxic snakes, there was a statistically significant association between EARs and the various types of hematotoxic antivenom that patients received. Despite the fact that GPV was the most common cause of snakebites and the most common type of snake for which antivenom therapy was required, patients with MPV bites who received antivenom had the highest incidence of EARs (37.8%). Therefore, the type of either hematotoxic snakebite or hematotoxic antivenom that patients receive might affect the risk of EARs. Further studies are needed to elucidate the mechanisms involved in this finding.

Most of the patients in our study received one type of antivenom, and EARs mostly occurred after the first dose of antivenom. Some patients received more than one dose of antivenom, and EARs also occurred in subsequent doses or intermittently, as demonstrated in one patient who received five doses of antivenom without any abnormal symptoms but developed EARs in the sixth dose. However, we found there was a higher incidence of EARs in those who received more than one type of antivenom (29.0%) than those who received a single type (22.3%). Accordingly, the type of antivenom of each dose that patients received and developed EARs was analyzed and calculated as the EAR rate. The overall EAR rate of all types of given antivenoms was 15%, and the three highest EAR rates were seen for the MPV, NPAV, and HPAV antivenoms. Thus, both monovalent and polyvalent antivenoms caused EARs. EARs to antivenom were generally mild to moderate, and cutaneous symptoms, mainly skin rash, were the most common reactions reported in our study. Respiratory symptoms were the second most frequent clinical manifestation of EARs. Both findings are consistent with previous studies of QSMI antivenoms [3,4,5,17,18,43]. Severe reactions occurred with an EAR rate of 2.6% and were mainly characterized by hypotension, which is the symptom most often reported in previous studies [5,11,17,18]. However, there were no fatalities associated with adverse events to antivenom. These findings will be helpful in planning appropriate treatment. Antivenoms with a high EAR rate and those that lead to severe EARs should be administered in well-equipped facilities with well-trained personnel so that prompt treatment can be given whenever anaphylactic reactions occur. However, there were no incidences of serum sickness or delayed antivenom reaction reported in this study; therefore, determining the incidence of delayed reactions was not included as a study objective. We concluded this might imply that serum sickness is not a common result of antivenom treatment.

The findings of a literature review of EARs associated with the use of QSMI antivenoms are summarized in Table 5. The incidence of EARs varied from no reactions being documented in two small retrospective studies [22,44] to an EAR incidence of 53% in one study [17]. A similar pattern of variation was found for the occurrence of severe reactions, as illustrated in Table 5. The overall incidence of EARs and severe EARs in the present study was higher than that of most previous reports, but lower than that of Vongphoumy et al.′s study in Laos [17]. Contaminants or impurities present in the antivenoms might not explain this discrepancy because the antivenoms were produced by the same manufacturer using the same production protocols [21,26]. However, there were several differences among the studies, such as the types of snakes implicated, the type and number of antivenoms used, the study design/protocol and study period, in addition to the patient populations. In addition, the main aim of most of these studies [3,4,5,22,43,44] was not to determine the frequency of EARs to antivenoms. These differences among the studies might have contributed to the different incidences of EARs reported, including those in our study.

To learn more about EARs, we further compared our study to two other studies that demonstrated a large discrepancy between the incidence of EARs and severe reactions to antivenoms (Table 6). In a previous Thai study [11], the most common types of antivenom used in this study were GPV antivenoms (83.5%), while GPV antivenom was used in our study for only 29.5%. The majority of antivenom used in the Laos study [17] was MPV antivenom (53.5%). According to our findings, we found that MPV antivenom had the highest EAR rate and most severe EARs among the QSMI antivenoms. In addition, the use of antivenom preparations prior to administration was disparate, and in the two Thai studies, antivenom was given at a low concentration, whereas in the Laos study, antivenom was administered in undiluted form. These differences may explain why the Laos study reported a much higher incidence of EARs and more severe EARs than the two Thai studies.

In our study, most EARs including severe reactions occurred during the first hour after the administration of antivenom was initiated; nevertheless, there have been reports of EARs occurred within 1 h after the completion of administration. Thus, patients need to be closely and continuously monitored for the occurrence of EARs for a minimum of 2 h after the initiation of antivenom infusion. We also discovered that the duration of antivenom administration was associated with the occurrence of EARs. Use of the recommended infusion time of 30–60 min in the present study significantly lowered the incidence of EARs. However, this recommendation is controversial. One study demonstrated the infusion rate was related to the occurrence of EARs after antivenom administration [45]. Other studies reported that the speed of dose infusion did not alter the risk of EARs. A comparative study of administration methods involving a 30-min infusion and bolus injection over 10 min found no correlation with the incidence of EARs [39]. In addition, a randomized comparison trial of two antivenom infusion rates used in the treatment of 198 adult patients with snake envenoming in Sri Lanka [46] showed no difference in the occurrence of severe reactions between those given a 20-min infusion (IQR; 20–25 min) (rapid) or 2-h infusion (IQR; 75–120 min) (slow). These reports contrast with our findings; however, an infusion rate of 30–60 min may not have been the most common method used in previous studies. Additionally, the discrepancy might be explained by differences in the study population, type of snakebite, or type of antivenom used. Thus, we propose that an infusion rate of 30–60 min is the optimal rate for antivenom administration; although further studies are needed to confirm this conclusion.

According to our study, all eight types of QSMI antivenom can initiate EARs. Moreover, both the incidence of EARs and the EAR rate were higher than 10% for all types of snakebite and antivenom used, except for RV bites. Even though most EARs were mild to moderate in severity, some severe reactions occurred. All patients who experienced reactions responded well to treatment, and no patients died from EARs. Therefore, the risks of EARs should be weighed against the proven benefits of antivenom therapy. Whenever antivenom is indicated for treatment, the patient should receive the antivenom to alleviate the impacts of systemic envenomation without delay. Based on results of this study, for every type and dose of antivenom administered, especially hematotoxic and MPV antivenoms, which had the highest EAR rate and most severe EARs, patients should be infused for 30–60 min in conjunction with close observation, comprehensive monitoring, preparation of resuscitation equipment, and prompt treatment when EARs occur from the beginning of antivenom administration until at least 1 h after the administration has finished.

Only a small number of patients received premedication in this study; thus, we did not analyze the performance of premedication on EAR prevention. However, this implies that premedication is not commonly practiced by physicians in Thailand. Because the incidence of EARs reported in our study was quite high, the premedication should be studied further to ascertain its benefit to prevent EARs.

## 6. Limitations

Our study had some limitations, listed as follows. First, it is not compulsory to report cases of snakebites to the RPC. Therefore, not all snakebite victims were referred to our institution, and it is possible that incidences of EARs and EAR rates might differ from the findings reported in our study. Second, the retrospective nature of the study may have resulted in missing, incomplete, or unclear data. Third, the diagnosis of snake envenomation was mainly based on the patients’ claim that they were bitten by a snake together with other supporting information, including the snake’s area of distribution, the patient’s local and systemic clinical features of envenomation, laboratory abnormalities, and the patient’s response to specific antivenom, as the venom in the patients’ blood or urine could not be analyzed to confirm the diagnosis. Finally, records on individual factors, such as underlying disease, history of allergy, and history of previous horse serum exposure, were not complete, and these factors might have contributed to the EARs that occurred. Further studies are warranted to ascertain if these individual factors contribute to EARs and to determine the EAR rate of MPV antivenom to confirm our findings.

## 7. Conclusions

The overall incidence of EARs and the EAR rate after administering F(ab’)_2_ QSMI antivenoms were 22.5% and 15%, respectively. The EAR rates following each type of antivenom were all greater than 10%, except for RV antivenom. Hematotoxic antivenom causes a high EAR rate and severe EARs, especially in a patient who was bitten by MPV and received MPV antivenom. There were no EAR-related deaths following antivenom administration in our study. Though EARs to antivenom are an inevitable risk of snakebite therapy, each patient should receive the indicated antivenom without any delay. All types and every dose of antivenom should be infused for 30–60 min. Careful monitoring, preparation of resuscitation equipment, and continuous clinical observation are crucial for at least 2 h after antivenom administration to detect EARs and provide prompt treatment when anaphylactic reactions occur.

## Figures and Tables

**Figure 1 toxins-14-00694-f001:**
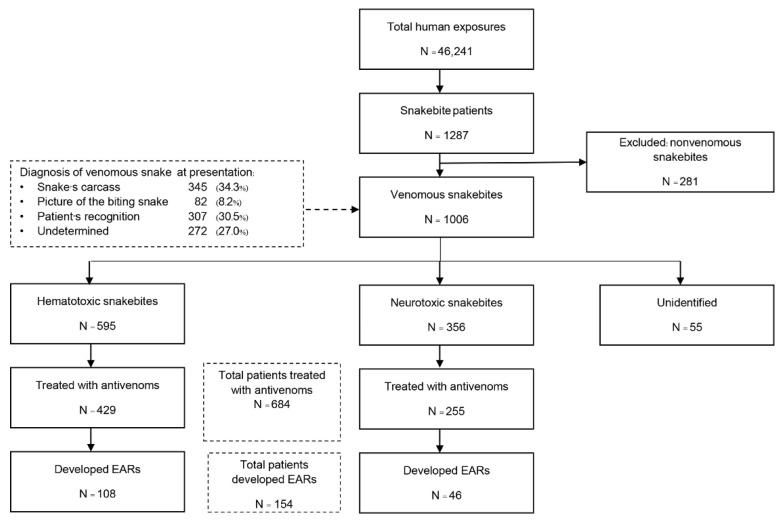
Flow diagram of study population.

**Figure 2 toxins-14-00694-f002:**
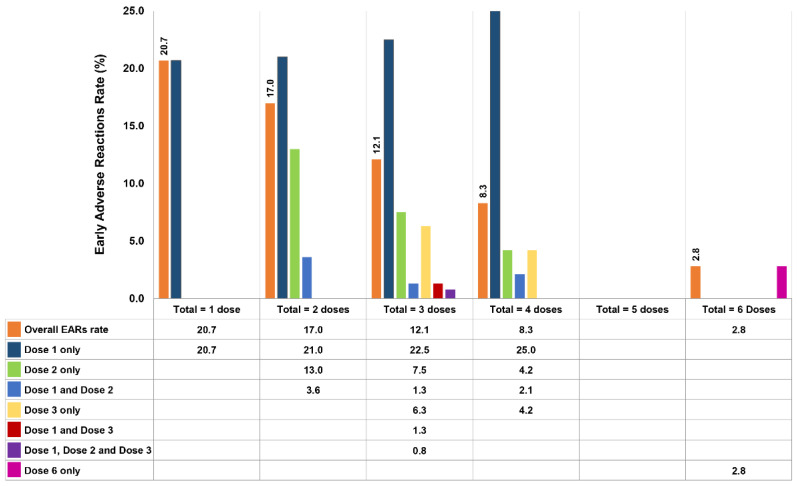
EAR rate (%) for each dose of antivenom in 154 patients (173 doses caused EARs out of 259 doses of antivenom in total).

**Figure 3 toxins-14-00694-f003:**
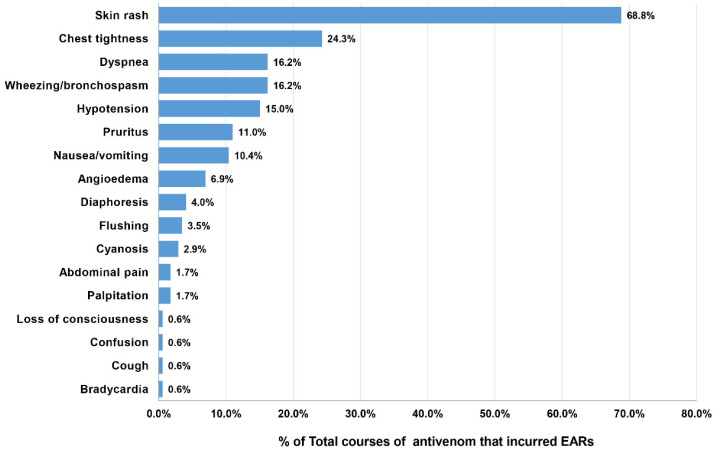
Reported clinical features of EARs in 154 patients after receiving 173 doses of antivenom.

**Table 1 toxins-14-00694-t001:** Characteristics of envenomed patients and snake species.

Characteristics	Total Venomous Snakebite Patients (N = 1006)
Age in years: Median (IQR, range)	38 (22–55, 0.6–97)
Gender ratio, Female:Male: N (%)	402:604 (40:60)
**VENOMOUS SNAKES CATEGORIZED BY SYSTEMIC ENVENOMING: N (%)**	
**Hematotoxic snakes, N = 595 (59** **.1%)**	
Green pit viper (*Trimeresurus* species)	336 (33.4%)
Malayan pit viper (*Calloselasma rhodostoma*)	168 (16.7%)
Russell’s viper (*Daboia siamenis*)	30 (3.0%)
Red-necked keelback (*Rhabdophis subminiatus*)	10 (1.0%)
Mountain pit viper (*Ovophis monticola*)	1 (0.1%)
Brown-spotted pit viper (*Protobothrops mucrosquamatus*)	1 (0.1%)
Unknown hematotoxic snakes	49 (4.9%)
**Neurotoxic snakes, N = 356 (35.4%)**	
Cobra (*Naja* species)	271 (26.9%)
Malayan krait (*Bungarus candidus*)	25 (2.5%)
King cobra (*Ophiophagus Hannah*)	10 (1.0%)
Coral snakes (*Calliophis* species)	9 (0.9%)
Banded krait (*Bungarus fasciatus*)	2 (0.2%)
Unknown neurotoxic snakes	39 (3.9%)
**Unidentified venomous snakes, N = 55 (5.5%)**	

AV; antivenom, EARs; early adverse reactions, N; number, IQR; interquartile range.

**Table 2 toxins-14-00694-t002:** Characteristics of envenomed patients who received antivenom and incidences of early adverse reactions.

Characteristics	No. of Patients Receiving AV	No. of Patients Receiving AV with EARs	Incidence of EARs (%)
Age in years: Median (IQR, range)	38 (21–55, 1–97)	35 (19–50, 1–87)	
Gender ratio, Female:Male: N (%)	266:418 (38.9:61.1)	34.4:65.6 (53:101)	
**FREQUENCY OF AV USED PER PATIENT, Median (IQR, range):**			
**Hematotoxic envenomation** Total doses/patient Total vials/patient	1 (1–2, 1–15) 5 (3–9, 1–75)	2 (1–2.8, 1–6) 5.5 (3–9, 1–18)	
**Neurotoxic envenomation** Total doses/patient Total vials/patient	1 (1–1, 1–4) 10 (10–10, 1–30)	1 (1–1, 1–4) 10 (10–10,1–24)	
**VENOMOUS SNAKES CATEGORIZED BY SYSTEMIC ENVENOMING, N (%):**			
**Hematotoxic snakes:**	**^a^ 429 (62.7%)**	**^a^ 108 (70.1%)**	**25.2**
Green pit viper	^b^ 213 (31.1%)	^b^ 44 (28.6%)	*20.7*
Malayan pit viper	^b^ 143 (20.9%)	^b^ 54 (35.1%)	*37.8*
Russell’s viper	^b^ 28 (4.1%)	^b^ 2 (1.3%)	*7.1*
Red-necked keelback	1 (0.1%)	0	*0*
Mountain pit viper	1 (0.1%)	0	*0*
Unknown hematotoxic snakes	43 (6.3%)	8 (5.2%)	*18.6*
**Neurotoxic snakes:**	**255 (37.3%)**	**46 (29.9%)**	**18.0**
Cobra	188 (27.5%)	32 (20.8%)	*17.0*
Malayan krait	20 (2.9%)	4 (2.6%)	*20.0*
King cobra	7 (1.0%)	3 (1.9%)	*42.9*
Coral snakes	0	0	*0*
Banded krait	2 (0.3%)	1 (0.6%)	*50.0*
Unknown neurotoxic snakes	38 (5.6%)	6 (3.9%)	*15.8*
**Total**	**684 (100%)**	**154 (100%)**	**22.5**
**NUMBER OF TYPES OF ANTIVENOM THERAPY IN EACH PATIENT, N (%):**			
* One	651 (95.2%)	145 (94.2%)	**22.3**
** Two	31 (4.5%)	9 (5.8%)	**29.0**
*** Three	2 (0.3%)	0	**0**

AV; antivenom, EARs; early adverse reactions, N; number, IQR; interquartile range. * Only one specific antivenom was given, either monovalent or polyvalent antivenom. ** Two specific types of monovalent and/or polyvalent antivenoms were given. *** Three different types of specific monovalent and/or polyvalent antivenom were given. ^a^
*p*-value = 0.031, ^b^
*p*-value = 0.001.

**Table 3 toxins-14-00694-t003:** EAR rate of each antivenom and severity grading of EARs.

Types of AV Used (Quantity Given)	Frequency of AV Administration		EAR Rate (%)	Grading of EARs, N (%)		
	Total Doses Given N (%)	No. of Doses That Incurred EARs (N)		Mild	Moderate	Severe
**Hematotoxic AV:**						
GPV (3860 vials)	341 (29.5)	39	11.4	14 (4.1)	24 (7.0)	1 (0.3)
MPV (3071 vials)	283 (24.5)	63	22.3	24 (8.5)	24 (8.5)	15 (5.3)
RV (1019 vials)	68 (5.9)	2	2.9	0 (0)	0 (0)	2 (2.9)
HPAV (1516 vials)	146 (12.6)	22	15.1	7 (4.8)	10 (6.8)	5 (3.4)
***Total* (*9466 vials*)**	***838* (*72.5*)**	** *126* **	** *15.0* **	***45* (*5.4*)**	***58* (*6.9*)**	***23* (*2.7*)**
**Neurotoxic AV:**						
Cobra (2754 vials)	213 (18.4)	31	14.6	17 (8.0)	11 (5.2)	3 (1.4)
MK (400 vials)	26 (2.2)	3	11.5	2 (7.7)	1 (3.8)	0 (0)
KC (164 vials)	7 (0.6)	1	14.3	0 (0)	0 (0)	1 (14.3)
NPAV (888 vials)	73 (6.3)	12	16.4	6 (8.2)	3 (4.1)	3 (4.1)
***Total*** **(*4206 vials*)**	***319* (*27.5*)**	** *47* **	** *14.7* **	***25* (*7.8*)**	***15* (*4.7*)**	***7* (*2.2*)**
**Overall Total** **(13,672 vials)**	**1157 (100)**	**173**	**15.0**	**70 (6.1)**	**73 (6.3)**	**30 (2.6)**

AV; antivenom(s), EARs; early adverse reactions, N; number of doses of AV, GPV; green pit viper, MPV; Malayan pit viper, RV; Russell’s viper, HPAV; hemato polyvalent antivenom, MK; Malayan krait, KC; king cobra, NPAV; neuro polyvalent antivenom.

**Table 4 toxins-14-00694-t004:** Characteristics of antivenom administration with and without EARs.

Characteristics (Total 1157 Doses)	No. of Doses with EARs (N = 173)	No. of Doses without EARs (N = 984)	*p*-Value
***Type of antivenom: N* (%)**			** *0.897* **
Received antivenom for hematotoxic snakes	126 (72.8)	721 (72.4)	
Received antivenom for neurotoxic snakes	47 (27.2)	272 (27.6)	
***Antivenom given dose:**N* (%):**			** *0.585* **
Appropriate dose	165 (95.4)	931 (94.6)	
Lower than appropriate dose	8 (4.6)	47 (4.8)	
Higher than appropriate dose	0 (0)	6 (0.6)	
***Duration of IV antivenom administration*:**			***0.002* ***
30–60 min (recommended method)	158 (91.3)	954 (97.0)	
<30 min	10 (5.8)	16 (1.6)	
>60 min	5 (2.9)	14 (1.4)	

EARs; early adverse reactions, N; number of doses of antivenom, IV; intravenous. ***** Statistically significant.

**Table 5 toxins-14-00694-t005:** Summary of studies in the literature reporting EARs associated with QSMI antivenoms.

Reference (Study Period)	Total Snakebite Patients	Type of Antivenom Used	Incidence of EARs %	Severe EARs %
Fung et al. [43] (2000–2005)	192	Cobra, GPV, KC	0	0
Thiansookon et al. [11] (1997–2006)	382	Cobra, GPV, MPV, RV	3.5	1.6
Blessmann et al. [44] (2007–2008)	21	MPV	0	0
Vongphoumy et al. [17] (2007–2008)	158	Cobra, GPV, MK, MPV, HPAV, NPAV	53.0	30.0
Mong et al. [18] (2008–2015)	191	GPV	4.7	1.6
Tongpoo et al. [4] (2008–2016)	78	BK, Cobra, MK, NPAV	6.0	0
Kraisawat and Promwang [42] (2006–2017)	153	MPV, HPAV	8.0	0

EARs; early adverse reactions, BK; banded krait, GPV; green pit viper, HPAV; Hemato polyvalent antivenom, KC; king cobra, MK; Malayan krait, MPV; Malayan pit viper, NPAV; Neuro polyvalent antivenom, RV; Russell’s viper.

**Table 6 toxins-14-00694-t006:** Comparison of studies of snakebite patients who received QSMI antivenoms.

Characteristics	Previous [11] Thai Study (2008)	Laos Study [17] (2016)	Our Study
Study period (duration in years)	1997–2006 (10)	2013–2015 (3)	2016–2017 (2)
Study design	Retrospective	Prospective	Retrospective
Age in years: Median (range)	39.3 ± 16.3 (mean ± SD)	32 (1.5–80)	38 (0.6–97)
Gender: Male (%)	60	68	60
No. of patients receiving AV	254	43	684
No. of patients with EARs	9	23	154
Incidence of EARs/Severe reactions (%)	3.5/1.6	53/30	22.5/2.6
Type of AV given in each study (%)	GPV = 83.5% Cobra = 12.6% RV = 2.4% MPV = 1.6%	MPV = 53.5% GPV = 16.3% Cobra = 7.0% MK = 4.7% HPAV = 11.6% NPAV = 7.0%	GPV = 29.5% MPV = 24.5% Cobra = 18.4% HPAV = 12.6% NPAV = 6.3% RV = 5.9% MK = 2.2% KC = 0.6%
Method of administration	diluted 100 mL/dose IV drip in 1 h	Undiluted (diluent 10 mL/vial IV drip 50 mL/h	diluted 100 mL/dose IV drip in 1 h

AV; antivenom, EARs; early adverse reactions, BK; banded krait, GPV; green pit viper, HPAV; hemato polyvalent antivenom, KC; king cobra, MK; Malayan krait, MPV; Malayan pit viper, NPAV; neuro polyvalent antivenom, RV; Russell’s viper, IV; intravenous.

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
