# Peer review of "Early Adverse Reactions to Snake Antivenom: Poison Center Data Analysis"

_toxins, 2022, doi:10.3390/toxins14100694_

Round 1
Reviewer 1 Report
This is an interesting, retrospective study looking at adverse effects associate with antivenom use. It is curious why the particular years were chosen and perhaps not more recent data but none the less, interesting.
Speculations regarding the etiology of the reactions are interesting and could be expanded as could data on how these reactions were both mitigated and dealt with when occuring. Perhaps to give greater insight into best practices.
Author Response
This is an interesting, retrospective study looking at adverse effects associate with antivenom use.
- It is curious why the particular years were chosen and perhaps not more recent data but none the less, interesting.
Speculations regarding the etiology of the reactions are interesting and could be expanded as could data on how these reactions were both mitigated and dealt with when occuring. Perhaps to give greater insight into best practices.
Reply: Thank you very much for your review.
We started doing this study in 2015 and then we designed to collect the data for 2 years by selecting the study period from January 2016 to December 2017. These 2-year periods, we collected and verified the data of 1,006 patients bitten by venomous snakes and 684 patients received the antivenom with 1,157 doses. Because there were a lot of data to collate and analyze, so we finished doing this study including writing the manuscript this year.
Since the antivenoms produced by the QSMI, The Thai Red Cross in Bangkok, Thailand during the study period (2016-2017) were under the same process and protocol as the antivenoms produced after those 2 years and until now. Moreover, our poison center’s recommendations for management of snakebite, antivenom treatment and management of EARs during the study period (2016-2017) were still the same as current recommendations. Therefore, we believe and suppose that the patients’ data during the study period (2016-2017) would not be obviously different from the recent or current period (2018-now).

Reviewer 2 Report
The authors present a retrospective study of the clinical characteristics and incidence of early adverse reactions (EAR) to QSMI antivenoms as reported to the Ramathibodi Poison Center (RPC) Toxic Exposure Surveillance System, a poison center database, from January 2016 to December 2017.
This retrospective study took care to provide as much detail as possible to identify patient characteristics, the species of snake involved, and the type of antivenom administered with or without premedication to prevent EAR. Further, the number of vials administered prior to EAR were noted, and the effects of the rate of administration on EAR also recorded.
Several hundred patients received various antivenoms for the treatment of known hemotoxic or neurotoxic venoms.
The results were disturbing. An incidence of EAR >20% seemed commonplace, with symptoms varying from skin rash (the most common EAR) to a variety of serious medical conditions (cyanosis, wheezing, bradycardia, etc.) as documented in figure 3. No patient died.
Overall, the work is interesting and serves as a call to refinement of the antivenoms to decrease EAR. However, there are a variety of serious matters to be attended to by the authors.
1. Scientific prose. The very large number of typographical and grammatical errors make this work a very hard read. The authors need to obtain assistance from an primary English speaking colleague.
2. Data presentation. The first two tables are nearly unreadable and must be adjusted so that data is presented on one line only as seen in table 3. Tables 4 and 5 also need to be improved. The font size in figure 2 and 3 are very small and hard to read – especially figure 2.
3. Dataset. Why did the authors choose such an old dataset, and why only one year? I cannot imagine that ongoing refinement of antibody or medical approach to EAR are the same from 2021 to the present.
4. EAR pretreatment. Given the very high incidence of EAR with these antivenoms, would it not be mandatory to premedicate all patients prior to administration? I know that the authors found only a small fraction of patients with premedication to assess in their database, but isn’t the finding of a large % of EAR a major motivation to promote premedication?
5. Clinical outcomes. While the investigation concerns EAR, how did the patients fair after treatment? Did they require transfusion, mechanical ventilation, etc.? Providing a small amount of data to indicate that that antivenom adequately treated the envenomation would be of interest to the readership.
Author Response
Manuscript ID number: toxins-1904918
Title of paper: Early Adverse Reactions to Snake Antivenom: Poison Center Data Analysis
Reviewer Comments:
Reviewer 2
- Extensive editing of English language and style required
Reply: We have used Edanz's editing services. Suzanne Leech who is the native English speaker and Molecular Parasitology PhD, was the person who helped editing our manuscript. We have attached the Certificate of Editing with our revision.
The authors present a retrospective study of the clinical characteristics and incidence of early adverse reactions (EAR) to QSMI antivenoms as reported to the Ramathibodi Poison Center (RPC) Toxic Exposure Surveillance System, a poison center database, from January 2016 to December 2017.
This retrospective study took care to provide as much detail as possible to identify patient characteristics, the species of snake involved, and the type of antivenom administered with or without premedication to prevent EAR. Further, the number of vials administered prior to EAR were noted, and the effects of the rate of administration on EAR also recorded.
Several hundred patients received various antivenoms for the treatment of known hemotoxic or neurotoxic venoms.
The results were disturbing. An incidence of EAR >20% seemed commonplace, with symptoms varying from skin rash (the most common EAR) to a variety of serious medical conditions (cyanosis, wheezing, bradycardia, etc.) as documented in figure 3. No patient died.
Overall, the work is interesting and serves as a call to refinement of the antivenoms to decrease EAR. However, there are a variety of serious matters to be attended to by the authors.
- 1. Scientific prose. The very large number of typographical and grammatical errors make this work a very hard read. The authors need to obtain assistance from an primary English speaking colleague.
Reply: We have used Edanz's editing services. Suzanne Leech who is the native English speaker and PhD Molecular Parasitology, was the person who helped editing our manuscript. We have attached the Certificate of Editing with our revision.
- 2. Data presentation. The first two tables are nearly unreadable and must be adjusted so that data is presented on one line only as seen in table 3. Tables 4 and 5 also need to be improved. The font size in figure 2 and 3 are very small and hard to read – especially figure 2.
Reply: We have edited Table 3-5 and Figure 2-3 as the reviewers suggested.
- 3. Dataset. Why did the authors choose such an old dataset, and why only one year? I cannot imagine that ongoing refinement of antibody or medical approach to EAR are the same from 2021 to the present.
Reply: We started doing this study in 2015 and then we designed to collect the data for 2 years by selecting the study period from January 2016 to December 2017. These 2-year periods, we collected and verified the data of 1,006 patients bitten by venomous snakes and 684 patients received the antivenom with 1,157 doses. Because there were a lot of data to collate and analyze, so we finished doing this study including writing the manuscript this year.
Since the antivenoms produced by the QSMI, The Thai Red Cross in Bangkok, Thailand during the study period (2016-2017) were under the same process and protocol as the antivenoms produced after those 2 years and until now. Moreover, our poison center’s recommendations for management of snakebite, antivenom treatment and management of EARs during the study period (2016-2017) were still the same as current recommendations. Therefore, we believe and suppose that the patients’ data during the study period (2016-2017) would not be obviously different from the recent or current period (2018-now).
- 4. EAR pretreatment. Given the very high incidence of EAR with these antivenoms, would it not be mandatory to premedicate all patients prior to administration? I know that the authors found only a small fraction of patients with premedication to assess in their database, but isn’t the finding of a large % of EAR a major motivation to promote premedication?
Reply: In Thailand, it is not mandatory to give premedication to all envenomed patients prior to antivenoms administration. Thank you for your suggestion, we have added the sentence “Because the incidence of EARs reported in our study was quite high, the premedication should be studied further to ascertain its benefit to prevent EARs.” into the discussion part.
- 5. Clinical outcomes. While the investigation concerns EAR, how did the patients fair after treatment? Did they require transfusion, mechanical ventilation, etc.? Providing a small amount of data to indicate that that antivenom adequately treated the envenomation would be of interest to the readership.
Reply: We have edited as the reviewers suggested by adding information of the clinical outcomes as the mortality rate of patients who received the antivenom into the results section (3.1).
We did not collect the data on transfusion or mechanical ventilation because we only focused on the data related to EAR.
Moreover, we did the study mainly in the patients who received the antivenoms and did not collect the data of patients who did not receive ones. So, we could not compare the clinical outcomes between patients who received or did not receive the antivenoms to assess the efficacy of antivenom.

Reviewer 3 Report
OVERAL IMPRESSION: The manuscript has significant detail especially in relation to the production and types of antivenom which is repeated in the introduction and the study protocol. DUe information overload in the methodology, one struggles to navigate the actual study protocol. The term "incidence" is used to descibe a variety of parameters such as rates, percentages and numbers. There needs to be consitency with using "incidence" that is maintained throughout he manuscipt.
INTRODUCTION: I would suggest defining the term EAR early rather than later in the script.
METHODOLOGY: The methods used to identify snake species is as mentioned in (2) and (3) of the study site and population section is open to significant observer and reporting bias. It is well documented that layperson/ witnesses are unreliable in identifying snake species (35% only) while using a sundromic apporach to snake species ID can be challenging as identifying to the exact snake species (27% accuracy). The authors also do not consider cytotoxicity as a syndrome or envenomation group which would be prevalent amongst ten Viper species.
The criteria for identifying haemotoxic bites includes and INR > 1.2 - can the authors sunstantiate the evidence for this cut-off. They also use severe swelling and pain as a criteria for haemotoxicity , however at best this is only a surrogate marker for potential associated haemotoxicity. This needs to be clarified.
The inclusion criteria for snake species criteria are repeated (line 152-159) which is uneccssary.
The repeated doses for antivenom are randomly administered @ 6 hours for haemotoxic envenomings and 12 hours for neurotoxic bites (line 162). Can the authors substantiate these delays in the repeat dosing regimen with evidence ?
The description of antivenom doses throughout is rather confusing and it is difficult to make head or tail what a dose is (? 1 vial). The dose of antivenom does not appear to influence the rate of anaphylaxis in studies from African snake bite antivenoms - can the authors comment on their findings.
RESULTS: Table 1 is very busy and has too much information to process in one table. Suggest breaking this into 2 tables for calrity and ease of reading. The number and types of antivenom is difficult to follow. Again could the authors clarify the dose of antivenom vs the number of vials per patient? Are these not the same.
Table 2; The number of doses to induce an EAR needs to e clrified - from the table it would aopoear for example that 39 doses would be needed before an EAR is experinced - confusing.
FIGURE 1: It is difficult to make head or tail of fig 1. Does each coulour represent a tyoe of antivenonm ? Is the overall parameter in the graph the mean, average or % ? Figure 1 requires more clarity. An interesting result is that the lowest EAR rate was at an infusion rate of 30-60 min, yet at faster or slower rates the EAR rate was high - this is difficult to understand and needs substantial comment from the authors. Antivenom infusion rate does not appear to influence the EAR rate from most of the mainstream evidence.
DISCUSSION: Including Tables 4 +5 in the discussion is perhaps a step to far and overloads the discussion content. Covention seldom includes tables in a discussion (rather they should be avoided or at a stretch be included in the results section). It would be he;lpful if the authros could present a national protocol for using antivenom, since they describe using chlorpheniramine and corticosteroids as prophylaxis to reduce EARs despite no evidence to support their use.
SUMMARY: The manuscript is a heavy read and at times difficult to follow. I suggest refining the whole manuscript to more succinct, clear and to the point.
Author Response
Manuscript ID number: toxins-1904918
Title of paper: Early Adverse Reactions to Snake Antivenom: Poison Center Data Analysis
Reviewer Comments:
Reviewer 3
OVERAL IMPRESSION: The manuscript has significant detail especially in relation to the production and types of antivenom which is repeated in the introduction and the study protocol.
- DUe information overload in the methodology, one struggles to navigate the actual study protocol. The term "incidence" is used to descibe a variety of parameters such as rates, percentages and numbers. There needs to be consitency with using "incidence" that is maintained throughout he manuscipt.
Reply: We have edited as the reviewers suggested.
We defined and used the term “incidence” to describe the data on “patients” who developed EARs. We showed how to calculate in Methods section, 2.3. Study protocol part (the number of patients with EARs divided by the total number of patients who received antivenom x 100).
While each dose of the different type of antivenom that each patient received was defined by the EAR rate (the number of doses of antivenom causing EARs divided by the total doses of antivenom used x 100).
- INTRODUCTION: I would suggest defining the term EAR early rather than later in the script.
Reply: We have edited as the reviewers suggested.
- METHODOLOGY: The methods used to identify snake species is as mentioned in (2) and (3) of the study site and population section is open to significant observer and reporting bias. It is well documented that layperson/ witnesses are unreliable in identifying snake species (35% only) while using a sundromic apporach to snake species ID can be challenging as identifying to the exact snake species (27% accuracy).
Reply: We have described the limitation of the identification of snake in our study that might not be all correct or reliable in the limitations section as “Third, the diagnosis of snake envenomation was mainly based on the patients’ claim that they were bitten by a snake together with other supporting information, including the snake’s area of distribution, the patient’s local and systemic clinical features of envenomation, laboratory abnormalities, and the patient’s response to specific antivenom, as the venom in the patients’ blood or urine could not be analyzed to confirm the diagnosis.
- The authors also do not consider cytotoxicity as a syndrome or envenomation group which would be prevalent amongst ten Viper species.
Reply: We have considered cytotoxicity as a local effect for envenomation and added “such as local swelling or coagulopathy” to the sentence in Page 3 “(2) The patient’s or relative’s description of the morphology of the snake and/or the snake’s species name. Because this information is sometimes unreliable, snake identification relied on the patient’s recognition corresponding to the area of distribution/habitat of the snake species, the patient’s local and systemic clinical features of envenomation such as local swelling or coagulopathy, laboratory abnormalities, and the patient’s response to a specific antivenom;”
- The criteria for identifying haemotoxic bites includes and INR > 1.2 - can the authors sunstantiate the evidence for this cut-off. They also use severe swelling and pain as a criteria for haemotoxicity , however at best this is only a surrogate marker for potential associated haemotoxicity. This needs to be clarified.
Reply: In Thailand, as mentioned in the manuscript that “antivenom therapy is recommended for systemic envenoming based on the recommendations of the QSMI [ref 26] and The Thai Society of Clinical Toxicology [ref 27].”
This was supported by “Regional Office for South-East Asia, WHO. Guidelines for the Management of Snakebites, 2nd ed.; WHO Regional Office for South-East Asia: New Delhi, India, 2016, ISBN 978-929-022- 530-0.”
Page 109: Antivenom treatment is indicated if/when patients with proven/suspected snakebite develop one or more of the following signs. Systemic envenoming: haemostatic abnormalities [spontaneous systemic bleeding, coagulopathy (+ve non-clotting 20WBCT, INR >1.2, or prothrombin time >4-5 seconds longer than control), or thrombocytopenia;
Page 126: The International Normalized Ratio (INR) is the patient’s PT divided by the
laboratory control PT (normal range 0.8 - 1.2). An abnormal INR result, indicating coagulopathy, is 1.2 or above.
Page 129: Indications for antivenom
Antivenom treatment is recommended if and when a patient with proven or suspected snakebite develops one or more of the following signs: Systemic envenoming
- Haemostatic abnormalities: spontaneous systemic bleeding distant from the bite site (clinical), coagulopathy [+ve (non-clotting) 20WBCT or other laboratory tests such as INR >1.2 or patient’s prothrombin time >4-5 seconds longer than laboratory control value] or thrombocytopenia [<100 x 109/litre, or <100 000/cu mm, or (India) < 1.0 lakh per microlitre of blood)] (laboratory)
And the publication as follows:
“Pongpit J, Limpawittayakul P, Juntiang J, Akkawat B, Rojnuckarin P.The role of prothrombin time (PT) in evaluating green pit viper (Cryptelytrops sp) bitten patients. Trans R Soc Trop Med Hyg. 2012 Jul;106(7):415-8. doi: 10.1016/j.trstmh.2012.04.003. Epub 2012 May 22.”
that described and concluded that “A fibrinogen level below 1.0g/litre was used as the gold standard. PT with INR (>1.2) can be an alternative test for evaluation of coagulopathy in green pit viper bitten patients with potentially improved inter-laboratory standardization.
- The inclusion criteria for snake species criteria are repeated (line 152-159) which is uneccssary.
Reply: We have edited as the reviewers suggested.
- The repeated doses for antivenom are randomly administered @ 6 hours for haemotoxic envenomings and 12 hours for neurotoxic bites (line 162). Can the authors substantiate these delays in the repeat dosing regimen with evidence ?
Reply: We described in the manuscript that “antivenom therapy is recommended for systemic envenoming based on the recommendations of the QSMI [ref 26] and The Thai Society of Clinical Toxicology [ref 27]”.
The evidence for repeated doses at 6 hours for haemotoxic envenoming
This was supported by “Regional Office for South-East Asia, WHO. Guidelines for the Management of Snakebites, 2nd ed.; WHO Regional Office for South-East Asia: New Delhi, India, 2016, ISBN 978-929-022- 530-0.”
Page 111: After the first dose of antivenom, the initial dose should be repeated 6 hours later if blood remains incoagulable, or 1 hour later if spontaneous systemic bleeding continues, or neurotoxic or cardiovascular signs persist or deteriorate. However, further doses of antivenom have no proven value in paralysed patients who are being ventilated.
Page 142: Criteria for giving more antivenom
- Persistence or recurrence of blood incoagulability after 6 hr or of bleeding after 1-2 hr
The evidence for repeated doses at 12 hours for neurotoxic bites
We advice antivenom treatment based on the recommendations of the QSMI [ref 26] and The Thai Society of Clinical Toxicology and practice antivenom therapy [ref 27], which are the experts consensus recommendation for the management of neurotoxic bites.
- The description of antivenom doses throughout is rather confusing and it is difficult to make head or tail what a dose is (? 1 vial).
The dose of antivenom does not appear to influence the rate of anaphylaxis in studies from African snake bite antivenoms - can the authors comment on their findings.
Reply: We have described in the manuscript (Page 4) that “One dose of antivenom for hematotoxic envenoming is 3– 5 vials, and for neurotoxic envenoming, it is 5–10 vials”.
The results of “FREQUENCY of AV USED PER PATIENT, Median (IQR, range)” as shown in Table 2 (the new table, breaking from Table 1) were the quantity of antivenom given to one patient by counting the number of doses and vials separately. Some patients received only 1 vial in the first dose and developed EARs, therefore, the other vials were withheld. So, the total vials of antivenom that patients received in hematotoxic snake envenomation ranging from 1 to 75 vials.
We also discussed more that “There was no association between antivenom dose and anaphylaxis” and added the study “Giles T, Čačala SR, Wood D, Klopper J, Oosthuizen GV. “A retrospective study of antivenom-associated adverse reaction and anaphylaxis at Ngwelezana Hospital, South Africa. Toxicon. 2022 Oct 15;217:1-4. doi: 10.1016/j.toxicon.2022.07.008. Epub 2022 Jul 20.” as the reference 41 in the discussion part”.
- RESULTS: Table 1 is very busy and has too much information to process in one table. Suggest breaking this into 2 tables for calrity and ease of reading. The number and types of antivenom is difficult to follow. Again could the authors clarify the dose of antivenom vs the number of vials per patient? Are these not the same.
Reply: We have edited as the reviewers suggested by breaking Table 1 into 2 tables (Table 1 and 2) and we have already described and clarified in the manuscript (Page 4) that “One dose of antivenom for hematotoxic envenoming is 3– 5 vials, and for neurotoxic envenoming, it is 5–10 vials.”
- Table 2; The number of doses to induce an EAR needs to e clrified - from the table it would aopoear for example that 39 doses would be needed before an EAR is experinced - confusing.
Reply: We have edited as the reviewers suggested by changing “No. of doses to induce EARs” to “No. of doses that incurred EARs” instead.
- FIGURE 1: It is difficult to make head or tail of fig 1. Does each color represent a type of antivenonm ? Is the overall parameter in the graph the mean, average or % ? Figure 1 requires more clarity.
Reply: This question is supposed to be Figure 2, not Figure 1. As the reviewer suggested, we have edited both font size and color of bar chart. All parameters present as EAR rate in percentage, and each color represents EAR rate of each dose of given antivenom that incurred EARs.
- An interesting result is that the lowest EAR rate was at an infusion rate of 30-60 min, yet at faster or slower rates the EAR rate was high - this is difficult to understand and needs substantial comment from the authors. Antivenom infusion rate does not appear to influence the EAR rate from most of the mainstream evidence.
Reply: We have discussed about the infusion rate in the manuscript based on our study’s finding (Page 18)
“Use of the recommended infusion time of 30–60 minutes in the present study significantly lowered the incidence of EARs. However, this recommendation is controversial. One study demonstrated the infusion rate was related to the occurrence of EARs after antivenom administration [45]. Other studies reported that the speed of dose infusion did not alter the risk of EARs. A comparative study of administration methods involving a 30-minute infusion and bolus injection over 10 minutes found no correlation with the incidence of EARs [38]. In addition, a randomized comparison trial of two antivenom infusion rates used in the treatment of 198 adult patients with snake envenoming in Sri Lanka [46] showed no difference in the occurrence of severe reactions between those given a 20-minute infusion (IQR; 20–25 minutes) (rapid) or 2-hour infusion (IQR; 75–120 minutes) (slow). These reports contrast with our findings; however, an infusion rate of 30–60 minutes may not have been the most common method used in previous studies. Additionally, the discrepancy might be explained by differences in the study population, type of snakebite, or type of antivenom used. Thus, we propose that an infusion rate of 30–60 minutes is the optimal rate for antivenom administration; although further studies are needed to confirm this conclusion.”
- DISCUSSION: Including Tables 4 +5 in the discussion is perhaps a step to far and overloads the discussion content. Covention seldom includes tables in a discussion (rather they should be avoided or at a stretch be included in the results section). It would be he;lpful if the authros could present a national protocol for using antivenom, since they describe using chlorpheniramine and corticosteroids as prophylaxis to reduce EARs despite no evidence to support their use.
Reply: Because these 2 tables (Table 4 and 5) were not commented to be removed by the other 2 reviewers, so we still keep these 2 tables. However, we have edited Table 4 and 5 as the reviewers suggested.
In Thailand, currently there is no national standard management guideline for snakebite; however, the common routine practice including our poison center’s practice/advice recommend antivenom treatment for systemic envenoming following the recommendations of the QSMI and The Thai Society of Clinical Toxicology.
- SUMMARY: The manuscript is a heavy read and at times difficult to follow. I suggest refining the whole manuscript to more succinct, clear and to the point.
Reply: We have edited as the reviewers suggested.

Round 2
Reviewer 2 Report
No further comments.